# Large genome-wide association study identifies three novel risk variants for restless legs syndrome

Maria Didriksen ⓘ et al.#

Restless legs syndrome (RLS) is a common neurological sensorimotor disorder often described as an unpleasant sensation associated with an urge to move the legs. Here we report findings from a meta-analysis of genome-wide association studies of RLS including 480,982 Caucasians (cases = 10,257) and a follow up sample of 24,977 (cases = 6,651). We confirm 19 of the 20 previously reported RLS sequence variants at 19 loci and report three novel RLS associations; rs112716420-G (OR = 1.25, $P = 1.5 \times 10^{-18}$), rs10068599-T (OR = 1.09, $P = 6.9 \times 10^{-10}$) and rs10769894-A (OR = 0.90, $P = 9.4 \times 10^{-14}$). At four of the 22 RLS loci, cis-eQTL analysis indicates a causal impact on gene expression. Through polygenic risk score for RLS we extended prior epidemiological findings implicating obesity, smoking and high alcohol intake as risk factors for RLS. To improve our understanding, with the purpose of seeking better treatments, more genetics studies yielding deeper insights into the disease biology are needed.

---

#A list of authors and their affiliations appears at the end of the paper.

Restless legs syndrome (RLS) is a common sensorimotor disorder that is known to impact quality of life and health[1,2]. The prevalence ranges from 5 to 18.8% in European populations[3–5] with approximately 2 to 3% of the general population thought to benefit from medical treatments that ameliorate symptoms[5–7]. RLS symptoms include uncomfortable sensations predominantly localized in the legs that are experienced as pain in at least one-third of subjects, which elicit a strong urge to move for symptomatic relief. The symptoms increase in the evening and at night. Consequently, the onset and maintenance of sleep are negatively impacted in most RLS patients, which in turn, is thought to impair daytime cognition and mental well-being[8]. The majority of RLS patients experience involuntary leg movements at transitions to sleep, and during sleep (periodic leg movements in sleep (PLMS)). Many also have social activities and work productivity interrupted by RLS symptoms[2].

One of the underlying pathophysiological mechanisms of RLS involves impaired re-uptake of synaptic dopamine and reduced D2 receptor density, explaining why the disorder can sometimes be treated with dopamine-based therapies[9]. It is hypothesized that the re-uptake of synaptic dopamine is affected by brain iron level[9]. Supporting this, in RLS patients low brain iron has been found in the substantia nigra and the striatum, whose roles in regulating reward, motivation, and movement are well established[10–12].

Moreover, a variety of modifiable health and lifestyle risk factors that accompany or aggravate RLS have been reported, including obesity, smoking, high alcohol intake, and sedentary lifestyle[3,13]. The prevalence is greater in individuals with reduced iron reserves[14]. Even though iron supplementation can be effective in relieving symptoms, especially in patients with iron deficiency, there are currently limited treatment options for RLS[15,16], which also appears to be underdiagnosed[17]. Existing treatments address symptoms rather than the underlying cause of the disease. A fundamental reason for this is our relatively limited knowledge of the pathogenesis of the disorder. One way to increase our understanding of RLS is to expand knowledge of the genetic architecture of the disorder, which is complex and polygenic in nature[6]. Genome-wide association studies (GWAS) of

European ancestry populations have yielded 20 single nucleotide polymorphisms (SNPs) in 19 loci that associate with RLS[6,18–24].

The aim of the present study was to search for additional RLS-associated loci that might provide new insights into the disease pathophysiology and be useful in the discovery of new drugs or repurposing of existing drugs for RLS treatment. To this end, a meta-analysis of GWAS of RLS including 480,982 adults of European ancestry (recruited from Iceland, Denmark, United Kingdom (UK), Netherlands and the United States (USA)) was conducted. Following this, novel findings were tested for replication in two additional case-control sets of European ancestry, the EU-RLS-GENE and RBC-Omics cohorts. Subsequently, all cohorts were meta-analyzed. Finally, to search for traits associated with RLS, we calculated polygenic risk scores for RLS (RLS-PRS) for the UK Biobank subjects and tested associations between RLS-PRS and 12,075 traits (binary and quantitative). The UK Biobank is one of the largest and most widely used recourses for studying health and well-being. The biobank sample is population-based, and the 500,000 volunteer participants provide health information to approved researchers by allowing the UK Biobank to link to existing health records, such as those from general practice and hospitals[25,26]. This study confirms 19 of the 20 previously reported RLS sequence variants at 19 loci and identifies three novel RLS-associated variants. Cis-eQTL analysis indicates a potential causal impact on gene expression at four of the 22 RLS loci. Finally, investigating traits associated with polygenic risk score for RLS, this study confirms and adds to prior epidemiological findings by implicating among other factors obesity, smoking and high alcohol intake as lifestyle risk factors for RLS.

## Results

**Genome-wide association study: discovery and replication.** The discovery meta-analysis confirmed 19 of the 20 previously reported RLS variants[6] (Fig. 1 and Supplementary Tables 1–3). The remaining SNP, rs12962305-T, had an effect size that was significantly smaller than previously reported meta-analyses (Table 1). The P-values of association with five sequence variants, at loci not previously associated with RLS, were below $5 \times 10^{-8}$ in the discovery sample and were tested in a follow up sample, including the

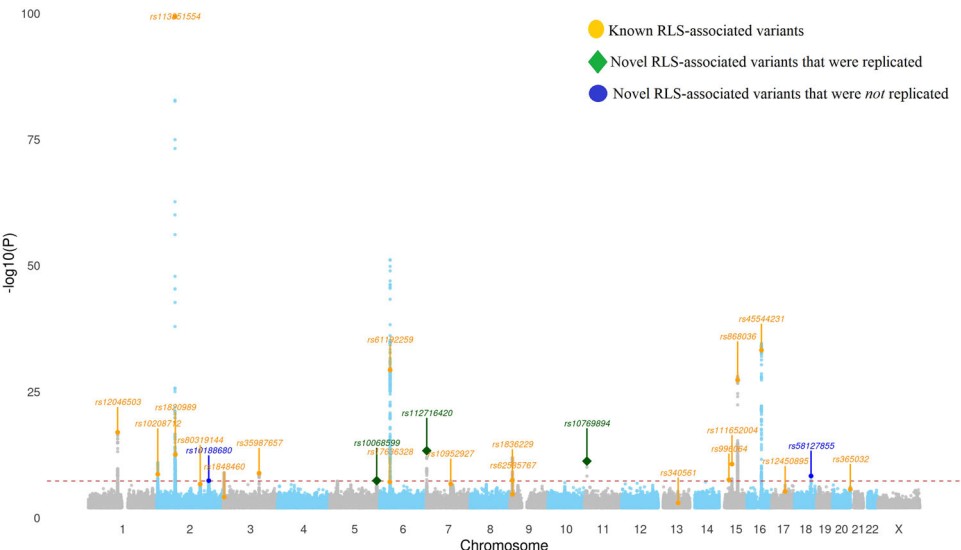

**Fig. 1 Manhattan plot displaying results from the RLS discovery meta-analysis for $N = 480,982$ independent biological samples.** Variants labeled orange are previously reported variants. Variants labeled blue and green are novel variants (five) that were tested in a follow-up sample. Of the five novel variants, three were confirmed (green diamond shape) in the follow up analysis and met the genome-wide significance threshold[27,28], whereas two did not (Table 1). (see Supplementary Table 1 for details; See Supplementary Figs. 1–5 for regional Manhattan plots displaying the five novel RLS-associated variants).

**Table 1 Sequence variants associated with RLS.**

| rsName | Chr | Position (hg38) | EA/OA | EAF | Genes | Discovery cases = 10,257 Controls = 470,725 OR (95% CI) | P | Follow up analysis[a] Cases = 6651 Controls = 18,326 OR (95% CI) | P | Combined analysis[b] Cases = 16,908 Controls = 489,051 OR (95% CI) | P |
|---|---|---|---|---|---|---|---|---|---|---|---|
| **Novel variants associated with RLS** | | | | | | | | | | | |
| rs10188680 | Chr2 | 189,584,800 | T/A | 0.41 | SLC40A1 | 1.09 (1.06-1.13) | $4.3 \times 10^{-08}$ | 1.04 (0.99-1.09) | 0.13 | 1.07 (1.05-1.11) | $5.4 \times 10^{-08}$ |
| rs10068599 | chr5 | 171,001,975 | T/C | 0.33 | RANBP17 | 1.10 (1.06-1.13) | $4.3 \times 10^{-08}$ | 1.07 (1.03-1.11) | $0.0031^c$ | 1.09 (1.06-1.12) | $6.9 \times 10^{-10}$ |
| rs112716420 | chr7 | 1,343,010 | G/C | 0.08 | MICALL2/UNCX | 1.24 (1.18-1.30) | $4.9 \times 10^{-14}$ | 1.27 (1.17-1.37) | $5.6 \times 10^{-06c}$ | 1.25 (1.19-1.31) | $1.5 \times 10^{-18}$ |
| rs10769894 | chr11 | 8,313,948 | A/G | 0.45 | LMO1 | 0.89 (0.86-0.93) | $5.8 \times 10^{-12}$ | 0.92 (0.87-0.97) | $0.0029^c$ | 0.90 (0.88-0.93) | $9.4 \times 10^{-14}$ |
| rs58127855 | Chr18 | 59,943,413 | T/C | 0.01 | PMAIP1 | 4.72 (4.20-5.24) | $5.1 \times 10^{-09}$ | 0.91 (−0.01-1.83) | 0.84 | 3.03 (2.01-4.97) | $6.3 \times 10^{-07}$ |
| **Known variants associated with RLS[d]** | | | | | | Current study Cases = 10,257 Controls = 470,725 | | Literature cases = 15,126 Controls = 95,725 | | Literature and current study combined Cases = 25,383 Controls = 566,450 | |
| rs10208712 | chr2 | 3,986,856 | G/A | 0.36 | . | 0.91 (0.88-0.94) | $2.34 \times 10^{-09}$ | 0.90 (0.87-0.93) | $3.78 \times 10^{-15}$ | 0.90 (0.88-0.92) | $5.9 \times 10^{-23}$ |
| rs0952927 | chr7 | 88,729,746 | G/A | 0.13 | | 1.13 (1.09-1.17) | $1.9 \times 10^{-09}$ | 1.17 (1.13-1.21) | $1.86 \times 10^{-15}$ | 1.15 (1.12-1.18) | $4.1 \times 10^{-21}$ |
| rs111652004 | chr15 | 47,068,169 | T/G | 0.10 | | 0.83 (0.77-0.88) | $2.2 \times 10^{-10}$ | 0.84 (0.79-0.89) | $1.05 \times 10^{-10}$ | 0.83 (0.79-0.87) | $1.5 \times 10^{-20}$ |
| rs113851554 | chr2 | 66,523,432 | T/G | 0.07 | MEIS1 | 1.89 (1.83-1.94) | $4.5 \times 10^{-100}$ | 2.16 (2.11-2.21) | $1.1 \times 10^{-180}$ | 2.03 (1.99-2.07) | $3.3 \times 10^{-276}$ |
| rs12046503 | chr1 | 106,652,717 | C/T | 0.41 | | 1.15 (1.11-1.18) | $1.09 \times 10^{-17}$ | 1.18 (1.15-1.20) | $3.32 \times 10^{-32}$ | 1.16 (1.14-1.18) | $7.1 \times 10^{-48}$ |
| rs12450895 | chr17 | 48,695,414 | A/G | 0.21 | | 1.09 (1.05-1.13) | $5.69 \times 10^{-06}$ | 1.09 (1.06-1.12) | $4.87 \times 10^{-08}$ | 1.09 (1.07-1.11) | $1.3 \times 10^{-12}$ |
| rs12962305 | chr18 | 44,290,278 | T/C | 0.25 | | 1.03 (1.01-1.05) | 0.0113 | 1.11 (1.08-1.14) | $1.37 \times 10^{-10}$ | 1.06 (1.04-1.08) | $4.5 \times 10^{-09}$ |
| rs17636328 | chr6 | 37,522,755 | G/A | 0.20 | | 0.90 (0.86-0.94) | $7.63 \times 10^{-08}$ | 0.89 (0.86-0.92) | $6.43 \times 10^{-11}$ | 0.89 (0.86-0.92) | $2.7 \times 10^{-17}$ |
| rs1820989 | chr2 | 67,842,758 | A/C | 0.47 | PTPRD | 1.12 (1.09-1.15) | $2.86 \times 10^{-13}$ | 1.14 (1.11-1.16) | $1.23 \times 10^{-20}$ | 1.13 (1.11-1.15) | $3.1 \times 10^{-32}$ |
| rs1836229 | chr9 | 8,820,573 | G/A | 0.48 | | 0.92 (0.89-0.95) | $3.68 \times 10^{-08}$ | 0.90 (0.87-0.93) | $1.94 \times 10^{-15}$ | 0.91 (0.89-0.93) | $6.2 \times 10^{-22}$ |
| rs1848460 | chr3 | 3,406,460 | T/A | 0.26 | | 1.06 (1.03-1.08) | $7.3 \times 10^{-05}$ | 1.13 (1.10-1.16) | $5.38 \times 10^{-14}$ | 1.09 (1.07-1.11) | $3.0 \times 10^{-15}$ |
| rs340561 | chr13 | 72,274,018 | T/G | 0.20 | | 1.07 (1.03-1.10) | 0.001 | 1.09 (1.06-1.12) | $3.93 \times 10^{-08}$ | 1.08 (1.06-1.10) | $2.5 \times 10^{-10}$ |
| rs35987657 | chr3 | 130,816,723 | G/A | 0.33 | | 0.90 (0.87-0.94) | $1.45 \times 10^{-09}$ | 0.90 (0.87-0.93) | $4.37 \times 10^{-13}$ | 0.90 (0.88-0.92) | $3.9 \times 10^{-21}$ |
| rs365032 | chr20 | 64,164,052 | G/A | 0.27 | MYT1 | 1.09 (1.05-1.12) | $2.13 \times 10^{-06}$ | 1.13 (1.10-1.16) | $3.36 \times 10^{-14}$ | 1.11 (1.09-1.13) | $1.5 \times 10^{-18}$ |
| rs45544231 | chr16 | 52,598,818 | G/C | 0.42 | | 0.82 (0.79-0.85) | $5.71 \times 10^{-34}$ | 0.81 (0.78-0.84) | $4.72 \times 10^{-48}$ | 0.81 (0.79-0.83) | $3.9 \times 10^{-80}$ |
| rs61192259 | chr6 | 38,486,186 | C/A | 0.41 | BTBD9 | 0.83 (0.80-0.86) | $4.71 \times 10^{-30}$ | 0.76 (0.73-0.79) | $1.36 \times 10^{-78}$ | 0.79 (0.77-0.81) | $1.9 \times 10^{-103}$ |
| rs62535767 | chr9 | 9,290,311 | T/C | 0.32 | PTPRD | 0.93 (0.89-0.96) | $2.2 \times 10^{-05}$ | 0.91 (0.88-0.94) | $3.13 \times 10^{-10}$ | 0.92 (0.89-0.94) | $4.8 \times 10^{-14}$ |
| rs80319144 | chr2 | 158,343,323 | T/C | 0.24 | CCDC148 | 0.91 (0.88-0.95) | $2.11 \times 10^{-07}$ | 0.89 (0.86-0.92) | $3.18 \times 10^{-14}$ | 0.90 (0.88-0.92) | $5.5 \times 10^{-20}$ |
| rs868036 | chr15 | 67,762,675 | T/A | 0.32 | MAP2K5 | 0.83 (0.79-0.86) | $4.67 \times 10^{-28}$ | 0.80 (0.77-0.83) | $1.09 \times 10^{-48}$ | 0.81(0.79-0.83) | $1.8 \times 10^{-74}$ |
| rs996064 | chr15 | 35,916,797 | T/A | 0.06 | | 1.21 (1.14-1.27) | $2.8 \times 10^{-08}$ | 1.21 (1.15-1.27) | $2.96 \times 10^{-09}$ | 1.21 (1.16-1.26) | $4.4 \times 10^{-16}$ |

EA is effect allele, OA is other allele, and EAF is effect allele frequency, OR is estimated odds ratio of the effect allele, P refers to association P-value of the tested allele, Gene closest gene with ±500kb.
[a]Follow up analysis of top five signals was carried out in two independent replication samples: EU-RLS-GENE cohort (cases/controls = 6228/10,992) and the RBC-Omics cohort (423/7334) (See Supplementary Table 1 for details and Supplementary Table 2, which displays results for all known RLS-associated variants).
[b]The combined analysis comprises both the discovery sample as well as the two replication samples.
[c]Represents significant P-value for replication samples after multiple testing: P < 0.05/5/2 = 0.005.
[d]Reference: PMID: 29029846.

EU-RLS-GENE cohort (6228 cases and 10,992 controls) and the RBC-Omics cohort (423 cases and 7,334 controls) (Supplementary Table 1 and Supplementary Figs. 1–5 for regional association plots). Three of the tested variants surpassed genome-wide significance in the meta-analysis of all samples[27,28] (Table 1). The novel RLS-associated sequence variants are; rs10068599-T in an intron of *RANBP17* on 5q35.1 (OR = 1.09, $P = 6.9 \times 10^{-10}$, 95% CI: 1.06–1.12), rs112716420-G in close proximity of *MICALL2* on 7p22.3 (OR 1.25, $P = 1.5 \times 10^{-18}$, 95% CI: 1.19–1.31) and rs10769894-A near *LMO1* and *STK33* on 11p15.4 (OR = 0.90, $P = 9.4 \times 10^{-14}$, 95% CI: 0.88–0.93) (Table 1).

**Cis-co-localization analysis of RLS variants using GTEx.** To identify the RLS variants acting as cis-expression quantitative trait loci (cis-eQTL) sharing the same signal with top eQTL of respective gene and tissue, we performed stepwise pairwise co-localization analysis. We investigated cis-eQTL of RLS variants in 54 tissues reported in the GTEx database. Of the 23 tested RLS variants (20 previously reported and three novel), we found cis-eQTL data for 11 impacting 17 genes (Supplementary Tables 4 and 5). Of the 11 with data, 10 strongly associate with cis-gene expression ($P < 3.3 \times 10^{-06}$, Supplementary Table 6). Six of these 10 variants are in LD ($r^2 > 0.3$) with top-eQTL for the respective gene (Supplementary Table 4). To ascertain that RLS variants and top-eQTLs share the same signal, we further evaluated these six variants by two-way approximate conditional analysis, which was implemented in COJO[29]. Therein, conditional analysis using RLS effect sizes showed that four RLS variants and eQTLs share the same signal (Supplementary Table 5). Additionally, conditional analysis using GTEx effect sizes also confirmed these as the same associated signals (Supplementary Table 6). Hence, four RLS variants (rs10068599-T, rs1063756-CACAG, rs12450895-A, and rs3784709-T) co-localize with top eQTLs for five genes respectively (*RANBP17, CASC16, HOXB2, MAP2K5,* and *SKOR1*) (Fig. 2) (for all RLS-associated variants see Supplementary Fig. 2).

rs10068599-T is associated with a lower expression of *RANBP17* in brain subcortical regions, mainly in the basal ganglia and in the liver, thyroid and heart left ventricle. rs3784709-T is associated with a lower expression of *SKOR1* in pituitary, pancreas, and mammary tissues, while the variant also is associated with a lower expression of *MAP2K5* in the left ventricle of the heart. Moreover, rs10653756-CACAG appears to be associated with a specific effect on *CASC16* expression in testes. rs12450895-A affects the expression of *HOXB2* by lowering it in suprapubic skin, fibroblasts cells, and in the omentum (visceral adipose tissue) (Fig. 2).

**Genetic risk and LD regression analysis.** We used RLS-PRS to predict RLS clinical cases ($N = 1916$ with the ICD10:G25.8 diagnostic code) in UK Biobank data. The analysis showed that RLS-PRS explains 0.97% of the phenotypic variance (Supplementary Fig. 7). One SD increase in RLS-PRS increases the odds of RLS 1.40-fold over that in population controls ($P = 4.4 \times 10^{-46}$, OR = 1.40, 95% CI: 1.35–1.45). Area under the curve and receiver operator curve analysis show that the risk for RLS increases for ascending quartiles (Supplementary Table 7 and Supplementary Fig. 8). RLS-PRS was used to identify traits associated with the score in the UK Biobank. Our analysis showed that higher RLS-PRS burden is negatively associated with educational attainment ($P = 2.7 \times 10^{-25}$, regression coefficient ($\beta$, continuous trait) = −0.02, standard error (SE): 0.002) and cognitive performance ($P = 4.4 \times 10^{-07}$, $\beta = -0.01$, SE: 0.002) and age at first time giving birth ($P = 5.9 \times 10^{-16}$, $\beta = -0.02$, SE: 0.003). The-PRS score furthermore associates positively with neuroticisms ($P = 8.0 \times 10^{-23}$, $\beta = 0.01$, SE: 0.002), as well as fat percentage in legs

($P = 1.4 \times 10^{-10}$, $\beta = 0.01$, SE: 0.002), and in the whole body ($P = 4.7 \times 10^{-07}$, $\beta = 0.008$, SE: 0.002) (Supplementary Tables 8 and 9). Results from LD score regression[30] and PRS-association analysis are in keeping (Supplementary Tables 10 and 11). The gene-set enrichment/pathway analysis using MAGMA[31] on a molecular signature database[32] recourse did not reveal any significant associations after correction for multiple testing (Supplementary Table 12).

## Discussion

Several sequence variants have been shown to associate with RLS, although causal variants at the associated loci and their functional relevance remains largely unknown. In a previous meta-analysis of RLS, 20 sequence variants at 19 loci were associated with RLS[6]. Here, we confirm associations with 19 of the 20 variants and report three novel associations bringing the number of RLS-associated variants to 23 at 22 loci. The three novel variants are rs112716420-G, rs10068599-T, and rs10769894-A.

The known protein-coding genes closest to rs112716420-G on chromosome 7 are *MICALL2* and *UNCX*. Variants in these genes are associated with red blood cell count and volume (i.e., hematocrit values), hemoglobin concentration and glomerular filtration rate[33–35]. rs112716420-G, however, does not associate significantly with these phenotypes in our samples. Hence, it does not appear that rs112716420-G impacts iron homeostasis, which is thought to be involved in the pathogenesis of RLS[11]. It is known that peripheral iron deficiency affects brain iron availability, although the specific mechanisms explaining how iron moves between the periphery and the nervous system remain unclear[9]. Moreover, the homeobox comprising transcription factor Uncx4.1 has been found to be expressed in glutamatergic, GABAergic and dopaminergic neurons in the mouse midbrain[36].

rs10068599-T is in an intron of *RANBP17 (Ran-binding protein 17)* on chromosome 5, which is a protein-coding gene of the exportin family. The cis-gene expression analysis showed that the rs10068599-T lowers the expression of *RANBP17* mainly in the basal ganglia and in the cerebral cortex. Previous studies have found that variants in *RANBP17* are associated with visceral fat[37], body mass index (BMI)[38], high-density lipoprotein (HDL) cholesterol[39], smoking status[40] and alcohol consumption[41].

The closest protein-coding gene to rs10769894-A on chromosome 11 is *LMO1*. This gene encodes the protein rhombotin-1, which is normally expressed in neural lineage cells[42,43]. Variants in *LMO1* have been associated with BMI[44] and neuroblastoma and T-cell leukemia[45,46], which is of interest since the strongest genetic predictor for RLS is a variant in *MEIS1* that affects cancers such as leukemia and neuroblastoma[47–49].

By integrating association statistics with gene expression data, we identified potential causal variants and genes affected at four of the 22 loci. As mentioned, the variant rs10068599-T lowers the expression of *RANBP17* in brain subcortical regions. rs3784709-T lowers the expression of *SKOR1* in pituitary, pancreas and mammary tissues. MEIS1 is considered an upstream activator of *SKOR1*[50], while rs12450895-A lowers the expression of *HOXB2* in adipose tissue and skin. Finally, we found that rs10653756-CACAG affects the expression of *CASC16* in testis. Hence, these variants may exert their causal effects through their impact on gene expression.

Our analysis showed that RLS-PRS, the aggregated genetic predisposition for RLS, correlates negatively with years of education and performance on cognitive tests but positively with neuroticism score. The RLS-PRS also correlates negatively with age at first birth and positively with several anthropometric measures, including whole body fat, percentage fat in trunk, legs and arms and waist-to-hip ratio. These findings extend prior

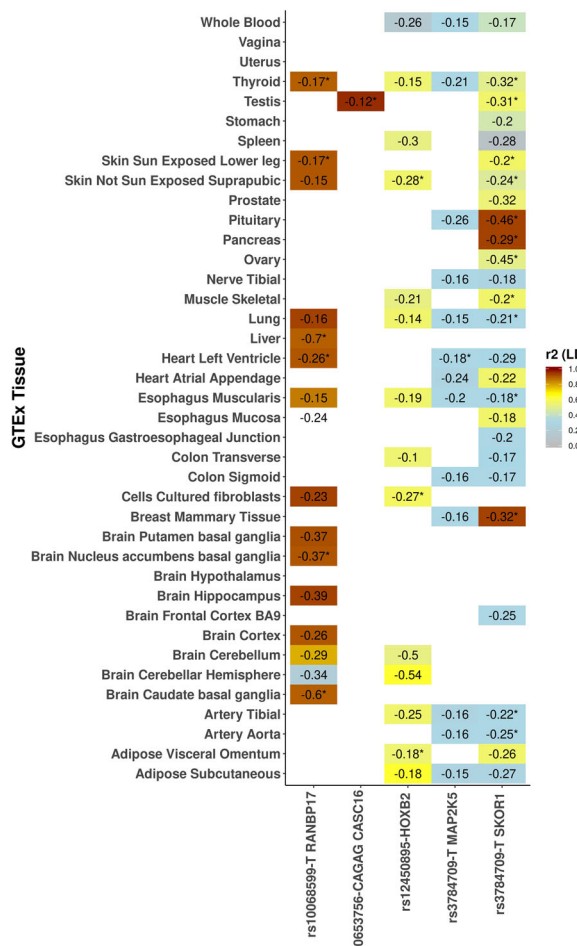

**Fig. 2 Cis-co-localization of RLS variants using 54 GTEx tissues. Displaying eQTL variants.** We found cis-eQTL data for 11 of the 23 RLS variants impacting 17 genes. Figure 2 displays the four variants that are significantly associated with cis-gene expression at least in one tissue tested are in linkage disequilibrium (LD) ($r^2 > 0.30$) and share the same causal signal (as confirmed through approximate conditional analysis) with the top eQTL variant of the respective genes (results for the remaining variants are displayed in Supplementary Fig. 6). Cis-eQTL effect estimates (normalized) are provided and those sharing same causal signal (COJO conditional analysis, results from this are displayed in Supplementary Table 5) with eQTL and are Bonferroni significant ($P < 3.3 \times 10^{-06}$) are labeled with an asterisk.

epidemiological studies[3] and both confirm and extend those of Schormair et al.[6] who searched for diseases and other traits associating with RLS-PRS. RLS has consistently been associated with modifiable lifestyles broadly considered to be unhealthy. In a prospective cohort study including 55,540 US adults, for example, RLS prevalence was lower among individuals who had a normal body weight, who were physically active, who were non-smokers, and who had an alcohol intake below the medium amount[13].

RLS is a complex polygenic sensorimotor disorder strongly influenced by lifestyle. This study increases the number of known independent RLS-associated genes to 23 in 22 loci, and cis-eQTL highlights genes at four of the loci giving more insights into RLS etiology. Future studies investigating the effect of drugs targeting the implicated physiological pathways and behavioral lifestyle changes on RLS as a therapeutic regime may provide valuable knowledge on the pathophysiology and the most prudent treatment modalities for RLS.

## Methods

**RLS status in the discovery samples.** The GWAS meta-analysis included 480,982 (10,257 cases and 470,725 controls) adults of European ancestry. Mean ages in included cohorts: Iceland 47.2 (SD, 14.06); Demark, 41.1 (SD, 12.3); the UK (Interval), 43.3 (SD, 14.1); the UK Biobank 60.0 (SD, 8.70); the Netherlands, 45.0 (14.0); and the US 56.5 (SD, 16.6). In total the analysis comprised 14,084 subjects from deCODE Genetics (Iceland) (2636 cases and 11,448 screened controls)[51], 26,565 subjects from The Danish Blood Donor Study (DBDS) (Denmark) (1379 cases)[52,53], 27,988 subjects from the INTERVAL study (UK) (3065 cases)[54], 408,565 subjects from the UK Biobank (UK) (1916 cases)[55], 2363 subjects from the Donor InSight-III cohort (The Netherlands) (565 cases)[56] and 1417 subjects from the Department of Neurology and Program in Sleep at Emory University (Emory cohort) (US) (696 cases) (Fig. 3).

We used clinical diagnosis or questionnaire data to assess RLS status in the participants, either applying questions based on the International RLS Study Group (IRLSSG) diagnostic criteria for RLS[57,58] or the Cambridge-Hopkins RLS questionnaire (CH-RLSq), which is also based on these criteria. Definite and probable RLS cases were combined into one group[59,60] (questionnaires are displayed in "Questionnaires used to assess RLS" on page 4 in Supplementary material). For subjects in the UK Biobank, the clinical diagnostic code ICD10: G25.8 was used to inform affectation status, whereas for the Emory cohort, gold standard diagnosis derived from face-to-face clinical evaluations by RLS specialists was used and the controls were determined for those lacking symptoms and signs associated with RLS.

**Discovery meta-analysis.** In total, we tested 15,838,848 sequence variants (1000 Genome phase 3 panel markers) for association with RLS (For a more detailed description of the included cohorts, see section "Cohorts included in the discovery meta-analysis" on page 2 in Supplementary material and section "Genotyping, imputation, and association analysis of cohorts included in the discovery meta-analysis" on page 7 for a detailed description of the methods). The GWAS results from the six cohorts (Iceland, Denmark, UK INTERVAL, UK Biobank, US Emory, and the Netherlands) were combined using a fixed effect inverse variance model[61] allowing different allele frequencies (of genotypes) in each populations, i.e., based on the effect estimates and standard error. Moreover, to control for a heterogenetic effect of the markers tested in the populations, we used a likelihood ratio test (Cochran's Q) and so evaluated their test statistics.

Before conducting the meta-analysis, variants in each dataset were mapped to NCBI Genome reference Consortium Build 38 (GRCh38) positions and matched to the Icelandic variants based on position and alleles. We included variants that were properly imputed in all datasets and which have a minor allele frequency >0.1% in more than one cohort. For the suggestive associations we used conventional genome-wide P-value threshold of $P < 5 \times 10^{-08}$ to find lead associations and to test those for replication. To claim a novel genome-wide association the sequence variants used in the meta-analysis ($N = 15,838,848$) were split into five classes based on their genome annotation and the weighted significance threshold for each class was used[28] (for QQ-plot see Supplementary Fig. 9, and for principal component analysis plots see Supplementary Figs. 10 and 11).

**Replication of novel variants.** Novel variants identified in the discovery phase of our study were tested for association in two replication datasets consisting of subjects of European ancestry, the EU-RLS-GENE consortium[6] (6228 cases and 10,992 controls) and the RBC-Omics cohort (423 cases and 7334 controls)[62]. In both replication tests, analyses were adjusted for age, sex, and the first 10 principal components of ancestry in a logistic regression model (For a more detailed description of the included cohorts, see section "Cohorts used for follow-up/replication analysis" on page 6 in Supplementary material) (Fig. 3). For the suggestive associations we used conventional genome-wide threshold ($P < 5 \times 10^{-08}$) to find lead associations, which were tested for replication. To claim a novel genome-wide association the sequence variants used in the meta-analysis ($n = 15,838,848$) were split into five classes based on their genome annotation, and the weighted significance threshold for each class was used[28].

**Gene expression.** We assessed cis-eQTL effects of the variants associated with RLS. RNA sequencing data from 54 human tissues was obtained from the Genotype-Tissue Expression (GTEx) portal[63]. We tested all genes in a one Mb window centered on the 23 variants. In total 15,153 tests were performed, and Bonferroni threshold was applied to the P-value. Therefore, $P < 0.05/15,153 = 3.3 \times 10^{-06}$ was considered statistically significant.

**Genetic risk.** To assess the impact conferred by the confluence of common RLS variants we calculated a RLS-PRS for each of the 500,000 UK Biobank subjects. The RLS-PRSs were calculated using summary statistics from a subset of the RLS-GWAS meta-analysis (UK participants from the INTERVAL and the UK Biobank excluded). Briefly, to generate the RLS-PRS for the UK Biobank sample we used 630,000 informative SNPs across the genome and constructed locus allele-specific weightings by applying LDpred to the summary data from the subset meta-analysis GWAS[64]. Constructing individual weightings, we were able to calculate an aggregated score of genetic susceptibility for RLS in all included individuals.

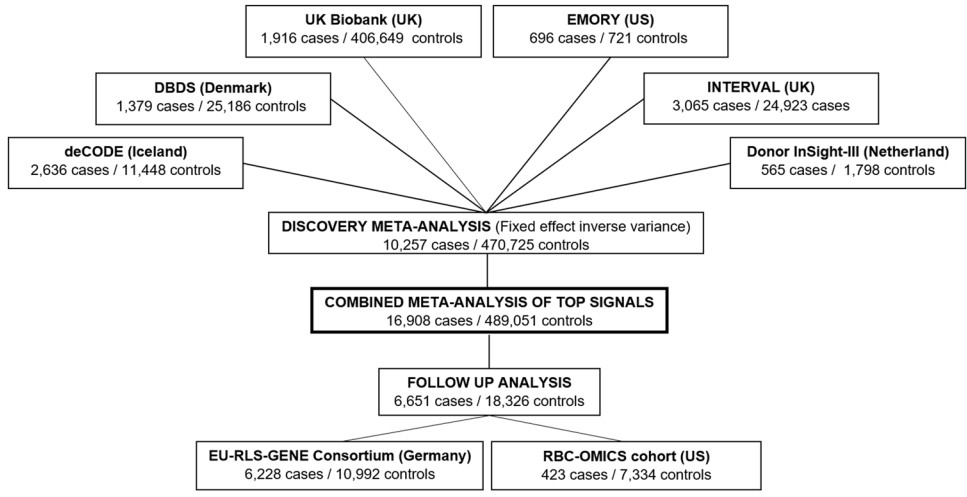

**Fig. 3 Overview of cohorts included in this study and the study scheme.** Displays the number of cases and controls of each cohort included in the present study—both in the Discovery meta-analysis ($N = 480{,}982$ independent biological samples), the follow-up analysis ($N = 24{,}977$ independent biological samples) and in the meta-analysis combining Discovery and Follow-up samples ($N = 505{,}959$ independent biological samples).

Subsequently, we assessed the impact of RLS-PRS on 12,075 traits (binary and quantitative) resulting in a Bonferroni significant threshold of $P < 0.05/12{,}075 = 4.14 \times 10^{-06}$.

**URLs**. GTEx, https://www.gtexportal.org/. The Genotype-Tissue Expression (GTEx).
COJO, https://cnsgenomics.com/software/gcta/#Overview.
SHAPEIT, https://mathgen.stats.ox.ac.uk/genetics_software/shapeit/shapeit.html.
PLINK2, https://www.cog-genomics.org/plink/2.0/
IMPUTE 2, https://mathgen.stats.ox.ac.uk/impute/impute_v2.html#download

**Ethics**. All sample identifiers were encrypted in accordance with the regulations of the Icelandic Data Protection Authority and written informed consent was collected from all study participants. The deCODE dataset was approved by the National Bioethics Committee of Iceland. The DBDS dataset was approved by The Scientific Ethical Committee of Central Denmark (M-20090237) and by the Danish Data Protection agency (30-0444). GWAS studies in DBDS were approved by the National Ethical Committee (NVK-1700407). The INTERVAL dataset was approved by the National Research Ethics Service Committee East of England - Cambridge East (Research Ethics Committee (REC: 11/EE/0538). The Emory dataset was approved by an institutional review board at Emory University, Atlanta, Georgia, US (HIC ID 133-98). The Donor InSight-III dataset was approved by the Medical Ethical Committee of the Academic Medical Center (AMC) in the Netherlands, and Sanquin's Ethical Advisory Board approved DIS-III and all participants gave their written informed consent. UK Biobank is approved by the North West Multi-center Research Ethics Committee, and by the Patient Information advisory Group, the National Information Governance Board for Health and Social Care, and from the Community Health Index Advisory Group. UK Biobank also holds a Human Tissue Authority license[65].

**Reporting summary**. Further information on research design is available in the Nature Research Reporting Summary linked to this article.

## Data availability
Data used in the present study is whole blood samples that have been genotyped. For this study, summary statistics from different RLS-GWAS's were collected and combined in a meta-analysis. The RLS meta-analysis summary statistics will be made available at https://www.decode.com/summarydata/. Data is available upon request. For access to data included in the meta-analysis, please contact the authors in charge of the respective cohorts. Henrik Ullum for data from the Danish Blood Donor Study (henrik.ullum@regionh.dk), Hreinn Stefansson for data from the Icelandic cohort (hreinn.stefansson@decode.is), David B. Rye for data from the Emory cohort (rlsrye@gmail.com), Emanuele Di Angelantonio for the INTERVAL cohort (ed303@medschl.cam.ac.uk), and Katja Van Den Hurk for data from the Donor Insight-III (k.vandenhurk@sanquin.nl). For UK Biobank please register on https://bbams.ndph.ox.ac.uk/ams/ and apply for the data through there.

## Code availability
Statistical codes are available upon request from corresponding author. No custom codes were used.

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

## Acknowledgements

We thank participants in all the included cohorts. More specifically, we thank the INTERVAL study coordination teams from Universities of Cambridge and Oxford, The NHS Blood and Transport (NHSBT), the 25 blood centers that recruited participants for this study, and the NIHR [Cambridge Biomedical Research Centre at the Cambridge University Hospitals NHS Foundation Trust] (The views expressed are those of the authors and not necessarily those of the NHS, the NIHR or the Department of Health and Social Care.). Similarly, we thank personnel employed in all blood banks across Denmark for making DBDS inclusion a part of their work routine. We wish to provide a thanks to Professor Richard P. Allen, Johns Hopkins University School of Medicine, Baltimore, US, for advice on identifying RLS cases among both the DBDS and the INTERVAL partici- pants. We also express our gratitude to Dr. Simone Glynn of NHLBI for her support throughout this study, to the RBC-Omics research staff at all participating blood centers and the testing laboratories. We thank the International EU-RLS-GENE consortium and the Cooperative Research in the Region of Augsburg (KORA) study for providing RLS-GWAS summary statistics. This study was supported with grants from The Danish Council of Independent Research—Medical Sciences (8018-00138A), The Danish Administrative Regions, The Danish Bio- and Genome bank, The Brothers Hartmann Foundation (A29659), Copenhagen University Hospital Research Foundation (Rig- shospitalets Forskningsfond). Participants in the INTERVAL randomized controlled trial were recruited with the active collaboration of NHS Blood and Transplant England (www. nhsbt.nhs.uk), which has supported field work and other elements of the trial. DNA extraction and genotyping was co-funded by the National Institute of Health Research (NIHR), the NIHR BioResource (http://bioresource.nihr.ac.uk/) and the NIHR Cambridge Biomedical Research Center (www.cambridge-brc.ac.uk). The academic coordinating center for INTERVAL was supported by core funding from: NIHR Blood and Transplant Research Unit in Donor Health and Genomics (NIHR BTRU-2014-10024), UK Medical

Research Council (MR/L003120/1), British Heart Foundation (RG/13/13/30194; RG/18/13/33946), and NIHR Cambridge BRC (a complete list of the investigators and contributors to the INTERVAL trial is provided in reference). The Emory cohort was made possible via funds provided by individual patients, and the Arthur L. Williams Jr. and Restless Legs Syndrome foundations. Moreover, the authors acknowledge NHLBI Recipient Epidemiology and Donor Evaluation Study-III (REDS-III), which was supported by NHLBI Contracts NHLBI HHSN2682011-00001I, -00002I, -00003I, -00004I, -00005I, -00006I, -00007I, -00008I, and -00009I. Finally, the UK Biobank was financed by the Wellcome Trust medical charity, Medical Research Council, Department of Health, Scottish Government, the Northwest Regional Development Agency, the Welsh Government, British Heart Foundation, and Cancer Research UK. The UK Biobank is supported by the National Health Service (NHS). The GTEx Project is available free-of-charge to researchers, and the project is funded by the Director of the National Institutes of Health, NCI, NHGRI, NHLBI, NIDA, NIMH, and NINDS. KORA was initiated and is financed by the Helmholtz Zentrum München, which is funded by the German Federal Ministry of Education and Research and by the state of Bavaria. The International EU-RLS-GENE consortium includes data from the COR study, which was supported by unrestricted grants to the University of Münster from the German Restless Legs Patient Organization (RLS Deutsche Restless Legs Vereinigung), the Swiss RLS Patient Association (Schweizerische Restless Legs Selbsthilfegruppe) and a consortium formed by Boeringer Ingelheim Pharma, Mundipharma Research, Neurobiotec, Roche Pharma, UCB (Germany + Switzerland) and Vifor Pharma. The clinical material and biospecimens of the Mayo Clinic Florida RLS collection were collected with the assistance of the Mayo Clinic internal funding through the Neuroscience Focused Research Team grant. Genotyping of the International EU-RLS-GENE consortium dataset was supported by DFG grant 218143125 to Prof. Juliane Winkelmann. The views expressed are those of the author(s) and not necessarily those of the NHS, the NIHR or the Department of Health and Social Care. *DBDS Genomic Consortium, responsible individuals*: Andersen Steffen, Department of Finance, Copenhagen Business School, Copenhagen, Denmark. Banasik Karina, Novo Nordisk Foundation Center for Protein Research, Faculty of Health and Medical Sciences, University of Copenhagen, Copenhagen, Denmark. Brunak Søren, Novo Nordisk Foundation Center for Protein Research, Faculty of Health and Medical Sciences, University of Copenhagen, Copenhagen, Denmark. Burgdorf Kristoffer, Department of Clinical Immunology, Copenhagen University Hospital, Copenhagen, Denmark. Erikstrup Christian, Department of Clinical Immunology, Aarhus University Hospital, Aarhus, Denmark. Hansen Thomas Folkmann, Danish Headache Center, department of Neurology Rigshospitalet, Glostrup. Hjalgrim Henrik, Department of Epidemiology Research, Statens Serum Institut, Copenhagen, Denmark. Jemec Gregor, Department of Clinical Medicine, Sealand University hospital, Roskilde, Denmark. Jennum Poul, Department of clinical neurophysiology at University of Copenhagen, Copenhagen, Denmark. Nielsen Kasper Rene, Department of Clinical Immunology, Aalborg University Hospital, Aalborg, Denmark. Nyegaard Mette, Department of Biomedicine, Aarhus University, Denmark. Paarup Helene Martina, Department of Clinical Immunology, Odense University Hospital, Odense, Denmark. Pedersen Ole Birger, Department of Clinical Immunology, Naestved Hospital, Naestved. Petersen Mikkel, Department of Clinical Immunology, Aarhus University Hospital, Aarhus. Sørensen Erik, Department of Clinical Immunology, Copenhagen University Hospital Copenhagen, Denmark. Ullum Henrik, Department of Clinical Immunology, Copenhagen University Hospital, Copenhagen, Denmark. Werge Thomas, Institute of Biological Psychiatry, Mental Health Center Sct. Hans, Copenhagen University Hospital, Roskilde, Denmark. Gudbjartsson Daniel, deCODE genetics, Reykjavik, Iceland. Stefansson Kari, deCODE genetics, Reykjavik, Iceland. Stefánsson Hreinn, deCODE genetics, Reykjavik, Iceland. Þorsteinsdóttir Unnur, deCODE genetics, Reykjavik, Iceland. *REDS-III, RBC-Omics STUDY GROUP MEMBERS:* The NHLBI Recipient Epidemiology Donor Evaluation Study- III (REDS-III), Red Blood Cell (RBC)-Omics Study, is the responsibility of the following Hubs: Blood Research Institute, Milwaukee, WI: A.E. Mast, J.L. Gottschall, W. Bialkowski, L. Anderson, J. Miller, A. Hall, Z. Udee, V. Johnson. The Institute for Transfusion Medicine (ITXM), Pittsburgh, PA: D.J. Triulzi, J.E. Kiss, P.A. D'Andrea. University of California, San Francisco, San Francisco, CA: E.L. Murphy, A.M. Guiltinan. American Red Cross Blood Services, Farmington, CT: R.G. Cable, B.R. Spencer, S.T. Johnson. Data coordinating center: RTI International, Rockville, MD: D.J. Brambilla, M.T. Sullivan, S.M. Endres-Dighe, G.P. Page, Y. Guo, N. Haywood, D. Ringer, B.C. Siege. Central and testing laboratories: Blood Systems Research Institute, San Francisco, CA: M.P. Busch, M.C. Lanteri, M. Stone, S. Keating. Pittsburgh Heart, Lung, Blood, and Vascular Medicine Institute, Division of Pulmonary, Allergy and Critical Care Medicine,

University of Pittsburgh, Pittsburgh, PA: T. Kanias, M. Gladwin. Steering committee chairman: University of British Columbia, Victoria, BC, Canada: S.H. Kleinman. National Heart, Lung, and Blood Institute, National Institutes of Health: S.A. Glynn, K.B. Malkin, A.M. Cristman. Finally, Muhammad S. Nawaz would also like to thank Marie Curie Initial Training Network for providing PhD funding through TS-EUROTRAIN grant (FP7-PEOPLE-2012-ITN, Grant Agr. No. 316978).

## Author contributions

M.D. and M.S.N. contributed to the conception, design, analysis, interpretation of data, and wrote first manuscript draft. J.D., S.B., D.B.R., L.M.T., K.V.D.H., F.Q., M.W.T.T., E.J.E., M.P.B., A.E.M., L.S., S.H.M., G.P.P., W.H.O., J.D., T.S., A.P.S., and D.J.R. contributed to acquisition, analysis, and interpretation of data and participated in revising the first manuscript draft. C.E., O.B.P., E.S., P.J.J., K.S.B., B.B., A.S.B., N.S., P.S., and G.T. participated in acquisition and interpretation of data and in revising the first manuscript draft. E.D.A., H.S., H.U., and K.S. contributed to the acquisition of data, conception and design of the study, analysis, interpretation of data, and to the revision of the first manuscript draft. All authors approved the submitted version of the manuscript. Finally, all authors are personally accountable for their own contributions, as well as the accuracy and integrity of any part of the work.

## Competing interests

A.E.M. received funding from Novo Nordisk. M.S.N., L.S., S.H.M., G.T., H.S., and K.S. are employees of deCODE genetics/Amgen. J.D reports grants, personal fees and non-financial support from Merck Sharp & Dohme (MSD), grants, personal fees and non-financial support from Novartis, grants from Pfizer and grants from AstraZeneca outside the submitted work. John Danesh sits on the International Cardiovascular and Metabolic Advisory Board for Novartis (since 2010); the Steering Committee of UK Biobank (since 2011); the MRC International Advisory Group (ING) member, London (since 2013); the MRC High Throughput Science 'Omics Panel Member, London (since 2013); the Scientific Advisory Committee for Sanofi (since 2013); the International Cardiovascular and Metabolism Research and Development Portfolio Committee for Novartis; and the Astra Zeneca Genomics Advisory Board (2018). A.B reports grants outside of this work from AstraZeneca, Biogen, BioMarin, Bioverativ, Merck, Novartis and Pfizer and personal fees from Novartis. The remaining authors have disclosed no conflicts of interest.

## Additional information

Maria Didriksen [1,2,25], Muhammad Sulaman Nawaz [2,3,25], Joseph Dowsett [1], Steven Bell [4,5,6], Christian Erikstrup [7], Ole B. Pedersen [8], Erik Sørensen[1], Poul J. Jennum[9,10], Kristoffer S. Burgdorf[1], Brendan Burchell [11], Adam S. Butterworth [4,5,6], Nicole Soranzo[4,12,13], David B. Rye[14], Lynn Marie Trotti[14], Prabhjot Saini[14], Lilja Stefansdottir[2], Sigurdur H. Magnusson[2], Gudmar Thorleifsson[2], Thordur Sigmundsson[3,15],

Albert P. Sigurdsson[3], Katja Van Den Hurk[16], Franke Quee[16], Michael W. T. Tanck[17],
Willem H. Ouwehand[4,12,13], David J. Roberts[4,18,19], Eric J. Earley[20], Michael P. Busch[21,22], Alan E. Mast[23],
Grier P. Page[24], John Danesh[4,5,6,13], Emanuele Di Angelantonio[4,5,6], Hreinn Stefansson[2],
Henrik Ullum[1,26✉] & Kari Stefansson[2,26]

[1]Department of Clinical Immunology, Copenhagen University Hospital, Rigshospitalet, 2100 Copenhagen, Denmark. [2]deCODE Genetics, 101 Reykjavik, Iceland. [3]Faculty of Medicine, University of Iceland, 101 Reykjavik, Iceland. [4]The National Institute for Health Research Blood and Transplant Unit in Donor Health and Genomics, University of Cambridge, Cambridge CB1 8RN, UK. [5]British Heart Foundation Cardiovascular Epidemiology Unit, Department of Public Health and Primary Care, University of Cambridge, Cambridge CB1 8RN, UK. [6]British Heart Foundation Centre of Research Excellence, Division of Cardiovascular Medicine, Addenbrooke's Hospital, Cambridge CB2 0QQ, UK. [7]Department of Clinical Immunology, Aarhus University Hospital, Aarhus, Denmark. [8]Department of Clinical Immunology, Nastved Sygehus, Nastved, Denmark. [9]Department of Clinical Neurophysiology, Danish Center for Sleep Medicine, Copenhagen University Hospital, Rigshospitalet, Glostrup, Denmark. [10]Faculty of Health, University of Copenhagen, Copenhagen, Denmark. [11]Faculty of Human, Social and Political Sciences, University of Cambridge, Cambridge CB1 8RN, UK. [12]Department of Haematology, University of Cambridge, Cambridge Biomedical Campus, Cambridge CB2 0PT, UK. [13]Department of Human Genetics, The Wellcome Trust Sanger Institute, Wellcome Trust Genome Campus, Hinxton, Cambridge CB10 1HH, UK. [14]Department of Neurology and Program in Sleep, Emory University, Atlanta, GA, USA. [15]Department of Psychiatry, Telemark Hospital Trust, Skien, Norway. [16]Department of Donor Studies, Sanquin Research, 1066 CX Amsterdam, The Netherlands. [17]Department of Clinical Epidemiology, Biostatistics and Bioinformatics, Amsterdam UMC, University of Amsterdam, Amsterdam, The Netherlands. [18]National Health Service (NHS) Blood and Transplant and Radcliffe Department of Medicine, NIHR Oxford Biomedical Research Centre, University of Oxford, John Radcliffe Hospital, Oxford, UK. [19]BRC Haematology Theme and Department of Haematology, Churchill Hospital, Oxford, UK. [20]RTI International, Research Triangle Park, Durham, NC, USA. [21]Vitalant Research Institute, San Francisco, CA, USA. [22]Department of Laboratory Medicine, University of San Francisco, San Francisco, CA, USA. [23]Blood Research Institute, Versiti, Milwaukee, WI, USA. [24]RTI International, Atlanta, GA, USA. [25]These authors contributed equally: Maria Didriksen, Muhammad Sulaman Nawaz. [26]These authors jointly supervised this work: Henrik Ullum, Kari Stefansson. ✉email: henrik.ullum@regionh.dk

