## [Peer Review File · Communications Biology]

Reviewers' comments:

Reviewer #1 (Remarks to the Author):

The manuscript is important in the field of restless legs genetics, well-designed and well-written. I have not seen any concern regarding its publication in the present form.

Reviewer #2 (Remarks to the Author):

In the current study, Didriksen et al. performed a genome-wide-meta analysis for RLS. Nineteen of the twenty known RLS variants were replicated, and three novel variants were reported. They performed the largest meta-analysis for RLS so far using samples from several cohorts and studies. Although it is a straightforward study with sound methodology and novel findings, some major points should be clarified.

Comments:

1- Prevalence of RLS is indicated as "from 5 to 10% in European population" in introduction and "as high as 10%" in abstract. Actually, prevalence is as high as 15% in several European ancestries. Ref no 3 is a study that was performed in a cohort of Danish blood donors. This population can be specified in the introduction. In ref no 5, many countries having a prevalence as high as 16-18% were reported. The prevalence should be clarified and cited according to the references. More references can be provided.

2- In supplementary table 1, the information for two variants rs10188680 and rs58127855 are relevant and it would be better to carry it to the results in the main text. The rs58127855 that has a Europe MAF < 0.1% is confusing. As low allele frequency variants were filtered and not included to the association test, how did this variant show up? It can also be clarified in results.

3- Have the authors considered performing a pathway or gene enrichment analysis? It would provide further insight into the mechanism implicated in RLS.

4- Would it be possible to draw a principal component plot at least for the discovery cohort using 1KG samples as a supplementary figure? Additionally, LocusZoom plots for the significant (especially novel) variants would be useful as a supplementary figure.

5- Legends and text should be provided more detail for all supplemental figures. For supplemental figure 2, the legend is very similar to the one for figure 3. The difference of the supplementary fig 2 and fig 3 should be clarified. For instance, in line 232, "lowering the expression of RANBP17 in several tissues", these tissues can be indicated.

-In supplementary materials, line 18, "...were recruited for the study (Supplementary Figure 1)." This should be corrected as (Figure 1).

-Figures should be numbered according to their order in the main text.

-Supplementary figure 3 should be cited in the main text. There is no information about it.

6- In line 116, the significance threshold should be included such as in parenthesis. In line 269, significance is defined as $P < 1.0 \times 10^{-8}$. How did the authors define this threshold?

7- In supplementary material, line 118, it is stated that "Samples were excluded if they were identified as ethnic outliers." How did the authors define the outliers here? Were π^2 or IBS scores used?

8- In the title and main text, it was stated that adults were included in this study. For UK Biobank,

it was indicated as "500,000 UK adults (ages 40 years and up)." For the other cohorts, what is the maximum or average age? If it was provided for other cohorts/studies, would it be possible to indicate in in methods?

9- In Methods, it is stated that "The UK Biobank is a most useful recourse for studying health and well-being. The biobank sample is population-based, and the 500,000 volunteer participants provide health information to approved researchers by allowing the UK Biobank to link to existing health records, such as those from general practice and hospitals." This part can be moved to introduction and should be cited. The UK Biobank can be one of the most useful or the largest resource, however it should be qualified with "to our knowledge", "so far" or equivalent phrase. Similarly, in line 57, novelty should be qualified with a phrase such as "to our knowledge".

10- It is hard to see the dbSNP ID annotations on the Manhattan plot. The font for dbSNP ID can be increased. Does chr 23 refer the chrX in the Manhattan plot? It should be clarified in the legend. Was this chromosome available in all the cohorts used in the meta analysis?

11- Citations or URLs should be added for the following: PLINK2 (line164), IMPUTE2 and 1000 Genome Project phase 3 (supp material, line 200). Line 392, SHAPIT should be corrected as SHAPEIT.

12- Abbreviations should be used in the first instance for HWE (supp material line 130 and line 157) and QC (supp material line 112 and line 115).

13- "19 of the 20 previously reported RLS variants". Which variant was not replicated? It would be easy to follow for the readers if the authors could indicate this variant in a sentence.

Reviewer #3 (Remarks to the Author):

In the paper by Didriksen et al., the authors have performed a meta-analysis of RLS GWASs and identified 3 novel risk loci associated with RLS.

Overall the paper is well written and interesting. One comment that I want to emphasize here (and detail below) is that the authors did not do a good job with the supplementary tables. We can't just throw the data into these tables, they need to be prepared with care and considering that they will be used by the readers of the paper.

Specific comments:

- Title: I don't see the importance of putting the number of participants in the title
- The study design is a bit odd, first a meta-analysis of several studies which the authors define as "discovery", and then a replication in other cohorts. Were all cohorts meta-analyzed eventually?
- When the authors mention "the combined sample" (line 116), do they refer to the replication cohorts combined? Or all cohorts combined? This should be written in a clearer way, including in the tables and supplementary tables.
- In line 116 the authors also refer to the new risk loci as "novel sequence variants". These are not novel sequence variants; they are known sequence variants. What's novel is the association with RLS. The authors should rephrase accordingly.
- If including in the text ORs and p values, 95% CI should also be included (for example lines 117-

119).

- Table 1 would benefit if it also included all the previously reported associations that were confirmed in the current study, so that we have a table with all the current known risk loci of RLS.

- Table 1 – is the effect allele frequency in %? Better to put allele frequencies (also in supp tables) as a 0-1 value, which is common practice in most GWASs.

- Supplementary Table 2: Where are the 95% CIs? What is the logic of putting OR and standard error? Either put betas and SE or put ORs and 95% CI. Please put MAF in the range of 0-1. Also, since the authors mention the effect alleles, wouldn't it be better to mention effect allele frequency rather than minor allele frequency (not always the same thing)? Otherwise we don't know if the allele frequency that they mention (the MAF) is the frequency of the effect allele or the other allele... What is the meaning of the column "max impact"? What is the meaning of the column "info"? Why are there no footnotes and abbreviations (explain for example what is Qp and I2 etc.)?

- Supp Table 3 – very confusing, a mess. Why are the variants not properly aligned in the rows of the table? For example, row 6 is rs10208712 in the beginning (column D), but then the same row refers to another variant, rs10177089 (Column L). The same is true for all rows. In addition, same comments as for supp table 2.

- Supplementary table 4 – again same comment, the authors mention MAF and then effect allele, but we don't know if the effect allele is always the minor allele? I am not sure about it. Better to mention effect allele frequency and effect allele, so that we are sure. Also, what is the logic behind the order of the variants in the different rows? It looks completely random, why not putting all the rows related to the same variant one after the other? For example, rs10068599 appears in row 6, then 8, then 11, 12 etc.

- I am stopping making comments on supplementary tables, as the issues are the same. Please prepare the supplementary tables properly, considering the above comments for all the tables.

- What do the authors mean by "causal signal" (line 143)? How do they know it is causal rather than associated?

- The authors write "rs10068599-T lowers the expression of RANBP17 in brain subcortical..", suggesting a causal relationship. We don't know that it is true. It is associated with lower expression, but we can't say it lowers the expression. It is not the same thing. The same comment for the following sentences as well (lines 148 – 154). These are all associations, and the authors cannot say things like "rs12450895-A affects the expression". To show this, they need to perform functional experiments, here there is only association between the SNP and expression levels.

- Why is it important to look at all these GTEx tissues, as many of them are very unlikely to be involved in RLS? (Fibroblasts? Heart? Etc.) Why is mentioning effects in "suprapubic skin" is relevant whatsoever?

- The order of the supplementary figures is incorrect, supp figure 2 appears in the text before 1.

- Please provide a detailed legend to supplementary figure 1. I have no idea what it means.

- Supplementary material – please improve it. Provide abbreviations so that readers don't have to go back and forth to the main paper. Words such as "affectation" are being used erroneously, how did the authors perform the harmonization of the genetic data?

- The authors say "The analysis showed that RLS-PRS explains 0.97% of the phenotypic variance (Supplemental Figure 1)." First, I don't see that in the figure, and I have no idea what the figure

means. Second, less than 1% of the phenotypic variance is extremely low, isn't it?

- It will be beneficial to perform ROC analysis and AUC with the PRS. In addition, it will be good to divide the individuals to groups (quartiles for example) and calculate for each group the relevant ORs.

- Again, missing 95% CI for the ORs mentioned in the PRS analysis.

- In the PRS analyses, what is the point mentioning in the text only the p values without the effect sizes?

- To determine associations with different traits, it would be useful to perform LD score regression, as well as bi-directional two-sample mendelian randomization to infer potential causality.

- No need to repeat in the discussion on p values and ORs, they are already mentioned before.

- I disagree with the sentence "By integrating association statistics with gene expression data, we identified likely causal variants and genes", mainly because of the word "likely". Replacing it with "potential" would be more appropriate. There is no proof here for likely causality.

- The authors should better describe how the harmonization of the genetic data was performed across the different cohorts, which used different SNP chips.

- How did the authors choose to use 10 principal components? Did they perform a Scree plot? They could be over fitting.

Reviewer #1 (Remarks to the Author):

The manuscript is important in the field of restless legs genetics, well-designed and well-written. I have not seen any concern regarding its publication in the present form.

Authors' response:

We are very happy with this feedback and we thank the reviewer for reading our manuscript.

Reviewer #2 (Remarks to the Author):

In the current study, Didriksen et al. performed a genome-wide-meta analysis for RLS. Nineteen of the twenty known RLS variants were replicated, and three novel variants were reported. They performed the largest meta-analysis for RLS so far using samples from several cohorts and studies. Although it is a straightforward study with sound methodology and novel findings, some major points should be clarified.

Authors' response:

Thank you for this very thorough review, which provided us with relevant constructive feedback. We believe that the reviewer's comments have helped to significantly improve our manuscript.

Comments:

1- Prevalence of RLS is indicated as "from 5 to 10% in European population" in introduction and "as high as 10%" in abstract. Actually, prevalence is as high as 15% in several European ancestries. Ref no 3 is a study that was performed in a cohort of Danish blood donors. This population can be specified in the introduction. In ref no 5, many countries having a prevalence as high as 16-18% were reported. The prevalence should be clarified and cited according to the references. More references can be provided.

Authors' response:

We appreciate the reviewer's point here and have updated the reported prevalence in both the abstract and manuscript.

Changes in abstract:

Following was added on page 3: "(...) s been reported as high as 18.8% (...)"

Changes in manuscript:

Following was added on page 4: "The prevalence ranges from 5 to 18.8% in (...)"

2- In supplementary table 1, the information for two variants rs10188680 and rs58127855 are relevant and it would be better to carry it to the results in the main text. The rs58127855 that has a Europe MAF < 0.1% is confusing. As low allele frequency variants were filtered and not included to the association test, how did this variant show up? It can also be clarified in results.

Authors' response:

We appreciate the reviewer's comment and have highlighted these two variants in Figure 2 (highlighted blue in the Manhattan plot). In addition to this, we included our findings on the two variants (rs10188680 and rs58127855) and on all known RLS-associated variants in Table 1 in the results section.

In the discovery meta-analysis, we retained all variants having MAF > 0.1% in more than one cohort. The rs58127855-T variant has variable allele frequencies in the different cohorts (MAF_{UKB} = 0.36%, MAF_{US} = 0.97%, MAF_{UK-INTERVAL} = 0.12%). Therefore, the variant was retained in the discovery meta-analysis.

Changes in manuscript:

On page 7 Figure 2 legend was revised: *"Figure 2: Manhattan plot displaying results from the RLS discovery meta-analysis (see Supplemental table 1 for details). Variants labeled orange are previously reported variants. Variants labeled blue and green are novel variants (five) that were tested in a follow-up sample. Of the five novel variants, three were confirmed (green diamond shape) in the follow up analysis and met the genome wide significance threshold^{27,28}, whereas two did not (Table 1). In the figure Chromosome 23 refers to chrX, chrX data was available for all cohorts except the US Emory. (See Supplemental figures 1 - 5 for regional Manhattan plots displaying the five novel RLS-associated variants)"*

On page 8: Data on known RLS-associated variants and on the five novel RLS-associated variants identified in the current study were added to Table 1

3- Have the authors considered performing a pathway or gene enrichment analysis? It would provide further insight into the mechanism implicated in RLS.

Authors' response:

We performed pathway/gene-set enrichment analysis using MAGMA, however our results did not add much to existing knowledge about RLS and therefore we did not include those results in the manuscript. We value the reviewer's suggestion and have therefore added a Supplemental Table 12, which display top pathway/gene-set terms from this analysis.

Changes in manuscript:

Following paragraph was added on page 13: *„LD score regression{Bulik-Sullivan, 2015 #22} analysis of the top binary and quantitative associations (identified in RLS-PRS analysis) suggests that the genetic correlations are in the same direction (though weaker than PRS) as observed in the PRS-analysis (Supplemental tables 10 and 11). The gene-set enrichment/pathway analysis using MAGMA {de Leeuw, 2015 #15} on a molecular signature database {Liberzon, 2015 #26} recourse did not reveal any significant associations (SupplementalTable 12).“*

Changes in Supplemental material:

Supplemental table 12 entitled 'Displaying top gene-set terms/pathways from MAGMA gene-set enrichment analysis using Molecular signature database resource.' was added.

4- Would it be possible to draw a principal component plot at least for the discovery cohort using 1KG samples as a supplementary figure? Additionally, LocusZoom plots for the significant (especially novel) variants would be useful as a supplementary figure.

Authors' response:

About PC plot:

We apologize for not clearly presenting meta-analysis process in the methods section. We have now tried to better explain the meta-analysis in the methods section.

Changes in Supplemental material:

Following paragraph was added on page 10: „ **Meta-analysis** Briefly, we did not perform a joint meta-analysis using complete data from all cohorts, instead we used summary statistics data for RLS association from Iceland, UK Biobank, UK INTERVAL, Denmark, US Emory, and the Netherland to run meta-analysis using fixed effect model (Mantel-Haenszel). The QC, imputation, and association tests, adjusting for principle components were done at cohort level and later meta-analyzed. Therefore, we have not done a joint principle component analysis of the discovery cohorts.

Detailed information about the QC, imputation, and the association method for Icelandic (Styrkarsdottir et al., 2019), UK Biobank (Astle et al., 2016), UK INTERVAL (Astle et al., 2016), and Danish (see methods) data used in this study have already been described by respective cohorts. For these samples we therefore believe that it is not necessary to generate the PC plots. However, for the US Emory, and the Dutch Insight-III cohorts we present information in the methods section including PCA plots showing genomic variation between cases and controls (below Supplemental Figure 10 and Supplemental Figure 11).“

Added references:

Astle, W. J., Elding, H., Jiang, T., Allen, D., Ruklisa, D., Mann, A. L., . . . Kostadima, M. A. (2016). The allelic landscape of human blood cell trait variation and links to common complex disease. *Cell*, 167(5), 1415-1429. e1419.

Styrkarsdottir, U., Stefansson, O. A., Gunnarsdottir, K., Thorleifsson, G., Lund, S. H., Stefansdottir, L., . . . Halldorsson, G. H. (2019). GWAS of bone size yields twelve loci that also affect height, BMD, osteoarthritis or fractures. *Nature communications*, 10(1), 1-13.

About LocusZoom plots:

We appreciate the valuable suggestion. The regional Manhattan plots (Locus zoom) for five novel sequence variants have been added as supplemental figures.

Changes in manuscript:

Following was added on page 7: „(See Supplemental figures 1 – 5 for regional Manhattan plots displaying the five novel RLS-associated variants)“

Changes in supplemental material:

LocusZoom plots were added on pages 14 to 18.

5- Legends and text should be provided more detail for all supplemental figures. For supplemental figure 2, the legend is very similar to the one for figure 3. The difference of the supplementary fig 2 and fig 3 should be clarified. For instance, in line 232, “lowering the expression of RANBP17 in several tissues”, these tissues can be indicated.

Authors’ response:

We agree with the reviewer’s observations. Therefore, we updated the numbers and legends for all supplemental figures and tables.

Changes in manuscript:

The legend was revised on page 9: *“We found cis-eQTL data for 11 of the 23 RLS variants impacting 17 genes. Figure 3 displays the four variants that are significantly associated with cis gene expression at least in one tissue tested are in linkage disequilibrium (LD) ($r^2 > 0.30$) and share the same causal signal (as confirmed through approximate conditional analysis) with the top eQTL variant of the respective genes (results for the remaining variants are displayed in Supplemental figure 6). Cis-eQTL effect estimates (normalized) are provided and those sharing same causal signal (COJO conditional analysis, results from this are displayed in Supplemental table 5) with eQTL and are Bonferroni significant ($P < 3.3 \times 10^{-06}$) are labeled with an asterisk.”*

Changes in supplemental material:

The legend was revised on page 18: **“Supplemental figure 6. Cis gene expression analysis of RLS variants using 54 GTEx tissues.** *Of the 23 RLS variants, we found cis-eQTL data for 11 of the 23 RLS variants impacting 17 genes. Four variants are significantly associated with cis gene expression at least in one tissue tested and are in high linkage disequilibrium (LD) ($r^2 > 0.90$) with the top eQTL variant of each respective gene. These are rs10068599-T lowering the expression of RANBP17 in the tissues thyroid, sun exposed skin on lower leg, liver, the left ventricle of the heart, and brain tissues in the basal ganglia. Moreover, the rs3784709-T which lowers the expression of SKOR1 and MAP2K5 in tissues pituitary, pancreas, and breast mammary tissue. Finally, the rs10653756-CACAG lowering the expression of CASC16 in testis. Cis-eQTL effect estimates (normalized) are provided and those sharing same causal signal (COJO conditional analysis) with eQTL and are Bonferroni significant ($P < 3.3 \times 10^{-06}$) are labeled with an asterisk.”*

A legend was added on page 16: **“Supplemental figure 7: The phenotypic variance explained by the RLS polygenic risk score (PRS) using different P-parameters from LDSC method.** *RLS-PRS for UKB participants were constructed using summary statistics from GWAS meta-analysis of Iceland, Denmark, US Emory, and the Netherland. Therein, the largest variance (0.97%) is explained by the PRS when applying the ‘0.01’ threshold for ICD 10 G25.8 in UKB. This PRS threshold (0.01) was further used to perform phenome-wide PRS association analysis of RLSPRS in UKB for 12,075 case-control (disease/phenotypes) and quantitative traits.”*

-In supplementary materials, line 18, “...were recruited for the study (Supplementary Figure 1).” This should be corrected as (Figure 1).

Authors’ response:

Thank you for noticing this error. This has been corrected.

- Figures should be numbered according to their order in the main text.
- Supplementary figure 3 should be cited in the main text. There is no information about it.

Authors' response:

We apologize for this and have corrected the figure numbers in the manuscript.

Changes in manuscript:

Page 17: *"For QQ-plot see supplemental figure 9."*

- 6- In line 116, the significance threshold should be included such as in parenthesis. In line 269, significance is defined as $P < 1.0 \times 10^{-8}$. How did the authors define this threshold?

Authors' response:

For the suggestive associations we used the conventional genome-wide threshold ($P < 5 \times 10^{-08}$) to find lead associations and to test those for replication. To claim a novel genome-wide association, the sequence variants used in the meta-analysis ($n = 15,838,848$) were split into five classes based on their genome annotation, and the weighted significance threshold for each class was used, as previously described⁵⁹.

Changes in manuscript:

Following paragraph was added in the Methods section on page 17: *"For the suggestive associations we used conventional GW threshold ($P < 5 \times 10^{-08}$) to find lead associations, which were tested for replication. To claim a novel GW association the sequence variants used in the meta-analysis ($n = 15,838,848$) were split into five classes based on their genome annotation, and the weighted significance threshold for each class was used, as previously described⁵⁹"*

- 7- In supplementary material, line 118, it is stated that "Samples were excluded if they were identified as ethnic outliers." How did the authors define the outliers here? Were pi hat or IBS scores used?

Authors' response:

This is a good comment. We have now elaborated on the exclusion process in Supplemental material.

Changes in supplemental material:

Following paragraph was added on page 8: *"We used the principal component analysis to identify and exclude outliers from the analysis. Briefly, to include samples in the study, we defined recent European ancestry (downloaded from <http://csg.sph.umich.edu/chaolong/LASER/>) as samples that fell into an ellipsoid spanning exclusively European population of the HGDP panel."*

- 8- In the title and main text, it was stated that adults were included in this study. For UK Biobank, it was indicated as "500,000 UK adults (ages 40 years and up)." For the other cohorts, what is the maximum or average age? If it was provided for other cohorts/studies, would it be possible to indicate in in methods?

Authors' response:

This is a nice comment as it will be beneficial for the reader to know this information. We have now added this information to both the cohort description in supplemental material and in the manuscript.

Changes in manuscript:

The following was added on page 13: *"The GWAS meta-analysis included 480,982 (10,257 cases and 470,725 controls) adults of European ancestry. Mean ages in the included cohorts: : Iceland 47.2 (S.D, 14.06); Demark, 41.1 (SD, 12.3); the UK (Interval), 43.3 (SD, 14.1); the UK Biobank 60.0 (S.D, 8.70); the Netherlands, 45.0 (14.0); and the US 56.5 (S.D, 16.6)."*

9- In Methods, it is stated that "The UK Biobank is a most useful recourse for studying health and well-being. The biobank sample is population-based, and the 500,000 volunteer participants provide health information to approved researchers by allowing the UK Biobank to link to existing health records, such as those from general practice and hospitals." This part can be moved to introduction and should be cited. The UK Biobank can be one of the most useful or the largest resource, however it should be qualified with "to our knowledge", "so far" or equivalent phrase. Similarly, in line 57, novelty should be qualified with a phrase such as "to our knowledge".

Authors' response:

We understand the reviewer's point and have changed the manuscript accordingly.

Changes in manuscript:

Page 5: The following paragraph was moved from the methods section to the introduction and the words "to our knowledge" were added: *"The UK Biobank is one of the largest and most widely used recourses for studying health and well-being. The biobank sample is population-based, and the 500,000 volunteer participants provide health information to approved researchers by allowing the UK Biobank to link to existing health records, such as those from general practice and hospitals."* With following references added:

- Bycroft, C. *et al.* The UK Biobank resource with deep phenotyping and genomic data. *Nature* 562, 203-209, doi:10.1038/s41586-018-0579-z (2018).
- Clare Bycroft, C. F., Desislava Petkova, Gavin Band, Lloyd T. Elliott, Kevin Sharp, Allan Motyer, Damjan Vukcevic, Olivier Delaneau, Jared O'Connell, Adrian Cortes, Samantha Welsh, Gil McVean, Stephen Leslie, Peter Donnelly, Jonathan Marchini. Genome-wide genetic data on ~500,000 UK Biobank participants. *BioRxiv*, doi:http://dx.doi.org/10.1101/166298 (2017).

Following paragraph was revised in the Abstract on page 3: *"The analysis confirmed 19 of the 20 previously reported RLS sequence variants at 19 loci and uncovered three novel RLS-associated variants; rs112716420-G (OR=1.25, P=1.5x10⁻¹⁸), rs10068599-T (OR=1.09, P=6.9x10⁻¹⁰) and rs10769894-A (OR=0.90, P=9.4x10⁻¹⁴)."*

10- It is hard to see the dbSNP ID annotations on the Manhattan plot. The font for dbSNP ID can be increased. Does chr 23 refer the chrX in the Manhattan plot? It should be clarified in the legend. Was this chromosome available in all the cohorts used in the meta analysis?

Authors' response:

Thank you for these valuable suggestions. We have added a higher resolution (with larger fonts) Manhattan plot. Yes, chr 23 refers to chrX. Except US Emory, the chrX summary statistics data was available for all cohorts in the discovery meta-analysis.

Changes in the manuscript:

In Figure 2 legend: *“In the figure Chromosome 23 refers to chrX, chrX data was available for all cohort except the US Emory”*

11- Citations or URLs should be added for the following: PLINK2 (line164), IMPUTE2 and 1000 Genome Project phase 3 (supp material, line 200). Line 392, SHAPIT should be corrected as SHAPEIT.

Authors’ response:

We have added the requested URLs.

Changes in the manuscript:

The following was added on Page 22:

„SHAPEIT, https://mathgen.stats.ox.ac.uk/genetics_software/shapeit/shapeit.html.

PLINK2, <https://www.cog-genomics.org/plink/2.0/>

IMPUTE 2, https://mathgen.stats.ox.ac.uk/impute/impute_v2.html#download“

Changes in the supplemental material:

The following was added on page 9: *„(...) the 1000 Genomes Phase 3 reference panel was used (URL, <https://www.internationalgenome.org/data-portal/data-collection/phase-3>).“*

12- Abbreviations should be used in the first instance for HWE (supp material line 130 and line 157) and QC (supp material line 112 and line 115).

Authors’ response:

Thank you for your keen observation. We have now added abbreviations in the revised supplemental material.

Changes in supplemental material:

Following was added on page 9: *“Hardy-Weinberg equilibrium (HWE)“*

On pages 10 and 11 „Hardy-Weinberg equilibrium“ was deleted

13- *“19 of the 20 previously reported RLS variants”*. Which variant was not replicated? It would be easy to follow for the readers if the authors could indicate this variant in a sentence.

Authors’ response:

We agree with the reviewer and we have now added this information to the results section of the manuscript.

Changes in the manuscript:

Following was added on page 6: *“The association of rs12962305-T with RLS was not confirmed in our meta-analysis. “*

Reviewer #3 (Remarks to the Author):

In the paper by Didriksen et al., the authors have performed a meta-analysis of RLS GWASs and identified 3 novel risk loci associated with RLS.

Overall the paper is well written and interesting. One comment that I want to emphasize here (and detail below) is that the authors did not do a good job with the supplementary tables. We can't just throw the data into these tables, they need to be prepared with care and considering that they will be used by the readers of the paper.

Authors' response:

Thank you for this great review of our manuscript. We agree that the Supplemental material needed a significant clean up. We carefully went through all the material and revised it according to the reviewer's comments. We believe that it is a lot better now and much easier to follow for the reader.

Specific comments:

1. Title: I don't see the importance of putting the number of participants in the title

Authors' response:

We agree with this point and have therefore changed the title of the manuscript to: *“Large genome-wide-association study identifies three novel risk variants for restless legs syndrome“*

2. The study design is a bit odd, first a meta-analysis of several studies which the authors define as “discovery”, and then a replication in other cohorts. Were all cohorts meta-analyzed eventually?

Authors' response:

We apologize for not clearly presenting the meta-analysis process in the methods section. We have now tried to provide a clearer explanation in the methods section.

Changes in the manuscript:

Following was added on page 5: *“Subsequently, all cohorts were meta-analyzed“*

Following was added on page 6: *“The discovery meta-analysis confirmed 19 of the 20 previously reported RLS variants⁶ (Figure 2, & Supplemental tables 1-3). The association of rs12962305-T with RLS was not confirmed in our meta-analysis. The P-values of association with five sequence variants, at loci not previously associated with RLS, were below 5×10^{-8} in the discovery sample and were tested in a follow up sample including the EU-RLS-GENE cohort (6,228 cases and 10,992 controls) and the RBC-Omics cohort (423 cases and 7,334 controls) (Supplemental table 1, Supplementary Figure 4-8 for regional association plots). Three of the tested variants met the significance threshold for genome wide association analysis^{27,28} in the combined sample, which included both the cohorts in the discovery meta-analysis as well as the replication cohorts (Table 1).“*

Changes in supplemental material:

Following was added on page 11: *“Briefly, we did not perform a joint meta-analysis using complete data from all cohorts, instead we used summary statistics data for RLS association from Iceland, UK Biobank, UK INTERVAL, Denmark, US Emory, and the Netherland to run meta-analysis using fixed effect model (Mantel-Haenszel). The QC, imputation, and association tests, adjusting for principle components were done at cohort level and later meta-analyzed. Therefore, we have not done a joint principle component analysis of the discovery cohorts.*

Detailed information about the QC, imputation, and the association method for Icelandic (Styrkarsdottir et al., 2019), UK Biobank (Astle et al., 2016), UK INTERVAL (Astle et al., 2016), and Danish (see methods) data used in this study have already been described by respective cohorts. For these samples we therefore believe that it is not necessary to generate the PCA plots. However, for the US Emory, and the Dutch Insight-III cohorts we present information in the methods section including PC plots showing genomic variation between cases and controls (below Supplemental Figure 10 and Supplemental Figure 11).”

3. When the authors mention “the combined sample” (line 116), do they refer to the replication cohorts combined? Or all cohorts combined? This should be written in a clearer way, including in the tables and supplementary tables.

Authors’ response:

We appreciate the reviewers point and have changed the manuscript accordingly.

Changes in manuscript:

The following was added the first time “combined sample” is written on page 6: *“the combined sample, which included both the cohorts in the discovery meta-analysis as well as the replication cohorts”*

The following was added in the legend of Table 1 on page 8: *„*Follow up analysis of top five signals was carried out in two independent replication samples: EU-RLS-GENE cohort (cases/controls=6,228/10,992) and the RBC-Omics cohort (423/7,334) (See Supplemental Table 1 for details)*

***The combined analysis comprises both the discovery sample as well as the two replication samples”*

Changes in supplemental material:

The following was changed in supplemental table 1:

- A title was added to supplemental table 1: *“Supplemental table 1. Displaying association results for the five novel RLS-associated variants identified in the discovery meta-analysis”*
- Following was added in supplemental table 1: *“Follow up (two replication cohorts: EU-RLS-GENE cohort and RBC-Omics cohort)”*
- 95% confidence intervals were added
- Following legend was added to the table:

*“P refers to P value for association, OR is odd ratio, 95% CI is 95% confidence interval
EAF is effect allele frequency.*

EAF-Europe is effect allele frequency in European populations.

Phet is P value for the tests of heterogeneity

I2 is percentage of variation across cohorts that is due to heterogeneity rather than chance.

EA is for effect-allele, OA is for other allele.

Info is imputation quality score, 0-1, i.e. Close to 1 mean more certain imputation.

S.E. is for standard error.”

The following was changed in supplemental table 2:

- A title was added to supplemental table 2: *“Supplemental table 2. Displaying association results for previously identified RLS-associated variants as well as association results for these variants in the discovery cohort from the present study as well as a meta-analysis of the results from the discovery cohort and previous results”*
- 95% confidence intervals were added

The following was changed in supplemental table 3:

- A title was added to supplemental table 3: *“Supplemental table 3. Displaying novel RLS-associated variants identified in the discovery meta-analysis as well as associations with known RLS-associated variants”*
- 95% confidence intervals were added

“P refers to P value for association, OR is odd ratio, 95% CI is 95% confidence interval, EAF is the effect allele frequency.

EAF-Europe is effect allele frequency in European populations.

Phet is P value for the tests of heterogeneity

I2 is percentage of variation across cohorts that is due to heterogeneity rather than chance.

EA is for effect-allele, OA is for other allele.

Info is imputation quality score, 0-1, i.e. Close to 1 mean more certain imputation.

S.E. is for standard error.”

Following explanatory legend was added to supplemental tables 2 and 3:

The following was changed in supplemental table 4:

- A title was added to supplemental table 4: *Supplemental table 4. Displaying genotype tissue expression for RLS-associated variants*
- A legend was added:

*“*is rsName for lead RLS variant*

eQTL is for top top eQTL in GTEx for the gene displayed in column G in the specific tissue displayed in column F.

r2: represents the degree of linkage disequilibrium of lead RLS-associated variant displayed in column A with GTEx top eQTL-variant for genes displayed in column G.

P refers to P value for association, OR is odd ratio, 95% CI is 95% confidence

interval, EAF is effect allele frequency.

EA is for effect-allele.

S.E. is for standard error.”

The following was changed in supplemental table 5:

- A title was added to supplemental table 5: „Displaying results from COJO conditional analysis of GTEx expression data for all variants with prior probability of P value < 3.3e-6 and degree of linkage disequilibrium ≥ 0.3 with the lead RLS-associated variants. analysis and therefore set Bonferroni threshold to P value < 2.01e-04“
- “ r^2 = Degree of linkage disequilibrium” was added in the table
- “Marker info” was changed to “Marker information”
- “Freq” was changed to “frequency of reference allele”
- “Freq_geno” was changed to “frequency of the genotype”
- Following legend was added to the table:

“**P** refers to P value for association, **Beta** is effect in standard deviation.

EA is for effect-allele, **OA** is for other allele.

S.E. is for standard error.”

The following was changed in supplemental table 6:

- A title was added to supplemental table 6: “Supplemental table 6. COJO conditional analysis of GTEx expression data for all variants with prior probability of P < 3.3e-6 and $r^2 \geq 0.3$ with the RLS lead variants. We performed 249 tests (220+29) for COJO conditional analysis and therefore set Bonferroni threshold to P < 2.01e-04 for different signal.”
- Following legend was added to the table:

“* If covariate is highly correlated ($r^2 \geq 0.9$) then COJO shows collinearity and therefore it's a same signal and no need to perform conditional association.

P refers to P value for association, **Beta** is effect in standard deviation.

EA is for effect-allele, **OA** is for other allele.

S.E. is for standard error.”

The following was changed in supplemental table 7:

- A title was added to supplemental table 7: „Supplemental table 7. Displaying results from association analysis between RLS-polygenic risk score and several binary health-related traits“
- 95% confidence intervals were added
- The columns “PRS” and “WT” were removed from the table as these do not bring valuable information to the reader

- “N affected” was changed to “N cases”
- All abbreviations in the “Phenotype” column were removed
- Following was added to column “covariates”: “Covariates included in association analysis”
- Following was added to the table: “PC=principal component” and “YOB=Year of birth”
- The table was sorted according to phenotype to improve overview of the presented data
- Following legend was added to the table:

“We tested 12,075 binary + quantitative traits (in UK Biobank sample) for association with polygenic risk score for RLS. Therefore we set Bonferroni significant threshold of $P < 0.05/12075$. The PRS was constructed using GWAS meta-analysis of RLS phenotype from Caucasian populations (Cases=8,558; Controls = 64,452).

PC=principal component

YOB=Year of birth

P refers to P value for association, OR is odd ratio, 95% CI is 95% confidence interval.

S.E. is for standard error.

R2 is variance explained in percentage.”

The following was changed in supplemental table 8:

- A title was added to supplemental table 8: „Supplemental table 8. Displaying results from association analysis between RLS-polygenic risk score and several quantitative health-related traits”
- Columns “PRS” and “WT” were deleted from the table because they did not bring valuable information to the reader
- Following was added to column “covariates”: “Covariates included in association analysis”
- Following was added to the table: “PC=principal component” and “YOB=Year of birth”
- The table was sorted according to phenotype to improve overview of the presented data

4. In line 116 the authors also refer to the new risk loci as “novel sequence variants”. These are not novel sequence variants; they are known sequence variants. What’s novel is the association with RLS. The authors should rephrase accordingly.

Authors’ response:

We agree with the reviewer on this and we have rephrased this paragraph accordingly.

Changes in manuscript:

Following was added on page 6: „The novel RLS-associated sequence variants are (...)”

5. If including in the text ORs and p values, 95% CI should also be included (for example lines 117-119).

Authors’ response:

We greatly appreciate this suggestion and we have now added the 95% confidence intervals.

Changes in manuscript:

95% CI’s were added in following paragraph on page 6: “The novel RLS-associated sequence variants

are; *rs10068599-T* in an intron of *RANBP17* (OR = 1.09, $P = 6.9 \times 10^{-10}$, 95% CI: 1.06 - 1.12) on 5q35.1, *rs112716420-G* in close proximity of *MICALL2* on 7p22.3 (OR 1.25, $P = 1.5 \times 10^{-18}$, 95% CI: 1.19 - 1.31) and *rs10769894-A* near *LMO1* and *STK33* on 11p15.4 (OR = 0.90, $P = 9.4 \times 10^{-14}$, 95% CI: 0.88 - 0.93)“

6. Table 1 would benefit if it also included all the previously reported associations that were confirmed in the current study, so that we have a table with all the current known risk loci of RLS.

Authors' response:

We understand the reviewer's point and we have therefore added this information to Table 1.

Changes in manuscript:

The following was added to Table 1 on page 8: Known RLS-associated variants (previous findings, findings on these variants in the current discovery cohort, and meta-analysis of previous findings combined with present findings on these variants)

7. Table 1 – is the effect allele frequency in %? Better to put allele frequencies (also in supp tables) as a 0-1 value, which is common practice in most GWASs.

Authors' response:

We thank the reviewer for suggesting this to keep a standard layout of GWAS summary statistics.

Changes in manuscript and supplemental material:

We have changed all reported allele frequencies to 0-1 values in both the manuscript and in supplemental materials.

8. Supplementary Table 2: Where are the 95% CIs? What is the logic of putting OR and standard error? Either put betas and SE or put ORs and 95% CI. Please put MAF in the range of 0-1. Also, since the authors mention the effect alleles, wouldn't it be better to mention effect allele frequency rather than minor allele frequency (not always the same thing)? Otherwise we don't know if the allele frequency that they mention (the MAF) is the frequency of the effect allele or the other allele... What is the meaning of the column "max impact"? What is the meaning of the column "info"? Why are there no footnotes and abbreviations (explain for example what is Qp and I2 etc.)

Authors' response:

We agree with the reviewer that more explanation should be added to the supplemental tables and we have now done so. Point 3 above addresses most of these. We also agree that it makes the most sense to report the allele frequency of the effect alleles. The effect alleles are the minor alleles, but this is not clear from the tables. We have now clarified this.

Max impact is a variant effect predictor, which is a variant detection and classification algorithm. The prediction is based on the location of the variant (in or near a gene), the predicted consequence of the variant on protein expression and the frequency of the variant in the human population.

Changes in supplemental material:

- We do not believe that the numbers in this column brings any useful knowledge to the readers and we have therefore removed the 'max impact' column from all tables.

- We have explained Qp and I2 in the tables that they occur (Please see reviewer comment 3. for a detailed description of this)
- In all tables: “MAF” was changed to “EAF”
- All reported allele frequencies were changed to a 0-1 range instead of a percentage.
- 95% confidence intervals have now been calculated and added to all relevant tables (Table 1, Supplementary tables 1, 2, 3 and 7).

9. Supp Table 3 – very confusing, a mess. Why are the variants not properly aligned in the rows of the table? For example, row 6 is rs10208712 in the beginning (column D), but then the same row refers to another variant, rs10177089 (Column L). The same is true for all rows. In addition, same comments as for supp table 2.

Authors’ response:

We really appreciate this comment and we have now corrected the error so that all rsNames are aligned in the table. Supplementary Table 3 has been thoroughly revised according to the reviewer’s comments. Please see Authors’ response to reviewer comment number 3 above as well as the revised version of Supplemental table 3.

10. Supplementary table 4 – again same comment, the authors mention MAF and then effect allele, but we don’t know if the effect allele is always the minor allele? I am not sure about it. Better to mention effect allele frequency and effect allele, so that we are sure. Also, what is the logic behind the order of the variants in the different rows? It looks completely random, why not putting all the rows related to the same variant one after the other? For example, rs10068599 appears in row 6, then 8, then 11, 12 etc.

Authors’ response:

Again, we really appreciate this comment and we agree. Regarding the comments about MAF vs. effect allele please see Authors’ response to comment number 8 above.

Changes in supplemental table 4:

We have sorted the table according to rsID’s to give the reader an easier overview of the data.

11. I am stopping making comments on supplementary tables, as the issues are the same. Please prepare the supplementary tables properly, considering the above comments for all the tables.

Authors’ response:

We have carefully revised all supplemental tables with the reviewer’s helpful comments in mind, and we feel that the supplemental material has been improved significantly.

12. What do the authors mean by “causal signal” (line 143)? How do they know it is causal rather than associated?

Authors’ response:

We agree with this point and we have changed the wording.

Changes in manuscript on page 9: “causal” was changed to “associated”.

13. The authors write “rs10068599-T lowers the expression of RANBP17 in brain subcortical..”, suggesting a causal relationship. We don’t know that it is true. It is associated with lower expression, but we can’t say it lowers the expression. It is not the same thing. The same comment for the following sentences as well (lines 148 – 154). These are all associations, and the authors cannot say things like “rs12450895-A affects the expression”. To show this, they need to perform functional experiments, here there is only association between the SNP and expression levels.

Authors’ response:

This is a valuable comment, which we agree with. We have revised the results section in the manuscript accordingly.

Changes in manuscript:

Following paragraph was edited on page 9: *“rs10068599-T is associated with a lower expression of RANBP17 in brain subcortical regions, mainly in the basal ganglia and in the liver, thyroid and heart left ventricle. rs3784709-T is associated with a lower expression of SKOR1 in pituitary, pancreas, and mammary tissues, while the variant also is associated with a lower expression of MAP2K5 in the left ventricle of the heart. Moreover, rs10653756-CACAG appears to be associated with a specific effect on CASC16 expression in testes.”*

14. Why is it important to look at all these GTEx tissues, as many of them are very unlikely to be involved in RLS? (Fibroblasts? Heart? Etc.) Why is mentioning effects in “suprapubic skin” is relevant whatsoever?

Authors’ response:

Our rationale for doing this is that the RLS physiology is still largely unknown. Therefore, we investigated all available eQTL data from GTEx (for 54 tissues) to understand the expression pattern of RLS-associated variants and ensure that we do not miss any relevant tissues.

15. The order of the supplementary figures is incorrect, supp figure 2 appears in the text before 1.

Authors’ response:

We apologize for the figure number errors and thank you for pointing this out. We have corrected the numbers of the supplemental figures and tables. Moreover, we have added an extra figure displaying locusZoom plots to supplemental material.

16. Please provide a detailed legend to supplementary figure 1. I have no idea what it means.

Authors’ response:

Supplemental figure 2 has been renamed to supplemental figure 7 and we have added an explanatory legend.

Changes in Supplemental material:

Following legend was added to Supplemental figure 2 on page 21: “The phenotypic variance explained by the RLS polygenic risk score (PRS) using different P-parameters from LDSC method. RLS PRS for UKB Biobank participants was constructed using GWAS meta-analysis of Iceland, Denmark, US Emory, and the Netherland. Therein, the largest variance (0.97%) is explained by the ‘0.01’ threshold for ICD 10 G25.8 in UK Biobank. This PRS threshold (0.01) was further used to perform phenome-wide PRS association analysis of RLS PRS in UK Biobank for 12,075 case-control (disease/phenotypes) and quantitative traits.”

17. Supplementary material – please improve it. Provide abbreviations so that readers don’t have to go back and forth to the main paper. Words such as “affectation” are being used erroneously, how did the authors perform the harmonization of the genetic data?

Authors’ response:

We have revised both the Supplemental material document as well as the tables according to the reviewer’s comment. We believe that this has been very beneficial.

Changes in manuscript:

Following paragraph on page 17 was elaborated: “The GWAS results from the six cohorts (Iceland, Denmark, UK INTERVAL, UK Biobank, US Emory, and the Netherland) were combined using a fixed effect inverse variance model⁵⁸ allowing different allele frequencies (of genotypes) in each populations i.e. based on the effect estimates and standard error. Moreover, to control for a heterogenetic effect of the markers tested in the populations, we used a likelihood ratio test (Cochran’s Q) and so evaluated their test statistics.

Before conducting the meta-analysis, variants in each dataset were mapped to NCBI Genome reference Consortium Build 38 (GRCh38) positions and matched to the Icelandic variants based on position and alleles. We included variants that were properly imputed in all datasets and which have a minor allele frequency > 0.1% in more than one cohort. For the suggestive associations we used conventional genome-wide (GW) P-value threshold of $P < 5 \times 10^{-08}$ to find lead associations and to test those for replication. To claim a novel GW association the sequence variants used in the meta-analysis (N = 15,838,848) were split into five classes based on their genome annotation and the weighted significance threshold for each class was used, as previously described⁵⁹. For QQ-plot see supplemental figure 9.”

Changes in Supplemental material:

Following sentence was changed on page 3: „(...)was used to inform about case status of restless leg syndrome“. Thus, the word „affectation“ was removed.

18. The authors say “The analysis showed that RLS-PRS explains 0.97% of the phenotypic variance (Supplemental Figure 1).” First, I don’t see that in the figure, and I have no idea what the figure means. Second, less than 1% of the phenotypic variance is extremely low, isn’t it?

Authors’ response:

We apologize for the unclear figure legend. We have added a more detailed legend to the Supplementary figure 2 (Please see Authors' response to reviewer's comment number 16).

We agree with reviewer's observation that a phenotypic variance of < 1% is low, which is quite common for complex traits. Moreover, it additionally underlines the phenotypic heterogeneity and polygenic nature of the RLS phenotype. We expect that with a larger sample size, we may be able to tag more of the genetic variants associated with RLS. This would expectedly increase the predictive power of the phenotype.

19. It will be beneficial to perform ROC analysis and AUC with the PRS. In addition, it will be good to divide the individuals to groups (quartiles for example) and calculate for each group the relevant ORs.

Authors' response:

We appreciate the kind suggestions by the reviewer. For readers ease, we have also added AUC plots using PRS (Supplemental figure 8) and we included a table showing ORs for each quartile (Supplemental table 7).

Changes in supplemental material:

One page 23 a figure was added with the following legend: *"ROC curve showing sensitivity and specificity of RLS PRS vs corresponding ICD 10-code G25.8 phenotype in the UK Biobank, where A) is for all and B-E) are for 1st, 2nd, 3rd and 4th quartile respectively."*

20. Again, missing 95% CI for the ORs mentioned in the PRS analysis.

Authors' response:

Thank you again for the kind observations, we have added 95% confidence intervals in all relevant tables and in the manuscript text.

Changes in the manuscript:

Odds ratios and 95% confidence intervals were added for binary health-related traits, while beta-values and standard errors were added for continuous health-related traits on page 11: *"One SD increase in RLS-PRS increases the risk of RLS 1.40-fold over that in population controls ($P = 4.4 \times 10^{-46}$, OR = 1.40, 95% CI: 1.35-1.45). RLS-PRS was then used to identify traits associated with the score in the UK Biobank. Our analysis showed that higher RLS-PRS burden is negatively associated with educational attainment ($P = 2.7 \times 10^{-25}$, regression coefficient (β , continuous trait) = -0.02, standard error (SE): 0.002) and cognitive performance ($P = 4.4 \times 10^{-07}$, $\beta = -0.01$, SE: 0.002) and age at first time giving birth ($P = 5.9 \times 10^{-16}$, $\beta = -0.02$, SE: 0.003). The-PRS score furthermore associates positively with neuroticisms ($P = 8.0 \times 10^{-23}$, $\beta = 0.01$, SE: 0.002), as well as fat percentage in legs ($P = 1.4 \times 10^{-10}$, $\beta = 0.01$, SE: 0.002), and in the whole body ($P = 4.7 \times 10^{-07}$, $\beta = 0.008$, SE: 0.002) (Supplemental tables 7 and 8)."*

Changes in supplemental material:

95% confidence intervals were added to supplemental tables 1, 2, 3, and 7.

21. In the PRS analyses, what is the point mentioning in the text only the p values without the effect sizes?

Authors' response:

We agree with the reviewer that it makes sense to include effect sizes as well. We have now added both odds ratios and 95% as well as beta-values and standard errors, respectively.

Changes in manuscript:

The following paragraph in the Results section on page 11 was edited: *“One SD increase in RLS-PRS increases the risk of RLS 1.40-fold over that in population controls ($P = 4.4 \times 10^{-46}$, OR = 1.40, 95% CI: 1.35-1.45). RLS-PRS was then used to identify traits associated with the score in the UK Biobank. Our analysis showed that higher RLS-PRS burden is negatively associated with educational attainment ($P = 2.7 \times 10^{-25}$, regression coefficient (β , continuous trait) = -0.02, standard error (SE): 0.002) and cognitive performance ($P = 4.4 \times 10^{-07}$, $\beta = -0.01$, SE: 0.002) and age at first time giving birth ($P = 5.9 \times 10^{-16}$, $\beta = -0.02$, SE: 0.003). The-PRS score furthermore associates positively with neuroticisms ($P = 8.0 \times 10^{-23}$, $\beta = 0.01$, SE: 0.002), as well as fat percentage in legs ($P = 1.4 \times 10^{-10}$, $\beta = 0.01$, SE: 0.002), and in the whole body ($P = 4.7 \times 10^{-07}$, $\beta = 0.008$, SE: 0.002)”*

Changes in supplemental material:

95% confidence intervals were added to supplemental tables 1, 2, 3, and 7.

22. To determine associations with different traits, it would be useful to perform LD score regression, as well as bi-directional two-sample mendelian randomization to infer potential causality.

Authors' response:

We appreciate the reviewer's suggestion and we have therefore performed LD score regression genetic correlation (GC) of RLS vs top associated phenotypes identified in the PRS analysis (Supplemental tables 10 and 11). The GC analysis did not show strong correlation between RLS and the respective phenotypes, but the correlations are in keeping with the PRS predictions. About MR analysis, we believe that it is not within the scope of this study as LDSC did not suggest a strong correlation requiring further dissection of the association at marker level.

23. No need to repeat in the discussion on p values and ORs, they are already mentioned before.

Authors' response:

The reviewer is correct on this point. We have therefore removed ORs and P values from the first paragraph of the discussion.

Changes in manuscript:

OR's and P values were removed from the following paragraph in the Discussion section on page 12: *“The three novel variants are rs112716420-G, rs10068599-T, and rs10769894-A.”*

24. I disagree with the sentence “By integrating association statistics with gene expression data, we identified likely causal variants and genes”, mainly because of the word “likely”. Replacing it with “potential” would be more appropriate. There is no proof here for likely causality.

Authors' response:

We agree with the reviewer and we have therefore moderated our wording in this sentence.

Changes in manuscript:

On page 13: "likely" was changed to "*potential*"

25. The authors should better describe how the harmonization of the genetic data was performed across the different cohorts, which used different SNP chips.

Authors' response:

Again, we apologize for not clearly explaining meta-analysis process. As stated above we did not perform a joint analysis using all cohorts, rather we meta-analyzed the GWAS summary statistics from six samples in the discovery meta-analysis. This has been explicitly explained in the Methods sections of both the manuscript and in supplemental material. Also, Figure 1 was revised to better present this.

26. How did the authors choose to use 10 principal components? Did they perform a Scree plot? They could be over fitting.

Authors' response:

We appreciate the reviewer's observations. In the follow up analysis, 10 principal components were used for in one cohort (US RB-Omics) to test five novel signals for replications. For this set, we were not concerned about over-fitting of the data but wanted to capture maximum variations in the population structure.